# Ascent and Attachment in Pea Plants: A Matter of Iteration

**DOI:** 10.3390/plants13101389

**Published:** 2024-05-16

**Authors:** Silvia Guerra, Giovanni Bruno, Andrea Spoto, Anna Panzeri, Qiuran Wang, Bianca Bonato, Valentina Simonetti, Umberto Castiello

**Affiliations:** Department of General Psychology, University of Padova, 35131 Padova, Italy

**Keywords:** climbing plants, kinematics, trial-and-error process, approach-to-grasp movement

## Abstract

Pea plants (*Pisum sativum* L.) can perceive the presence of potential supports in the environment and flexibly adapt their behavior to clasp them. How pea plants control and perfect this behavior during growth remains unexplored. Here, we attempt to fill this gap by studying the movement of the apex and the tendrils at different leaves using three-dimensional (3D) kinematical analysis. We hypothesized that plants accumulate information and resources through the circumnutation movements of each leaf. Information generates the kinematical coordinates for the final launch towards the potential support. Results suggest that developing a functional approach to grasp movement may involve an interactive trial and error process based on continuous cross-talk across leaves. This internal communication provides evidence that plants adopt plastic responses in a way that optimally corresponds to support search scenarios.

## Introduction

1

Climbing plants are characterized by a thin and flexible stem, which forces them to find potential environmental support to reach the greatest exposure to light. To clasp a support, climbing plants have evolved several morphological traits [1-3]. Among these, the most sophisticated are likely the tendrils [2]. Tendrils are modified leaves, stems, or flower peduncles sensitive to mechanical stimulation and are capable of coiling around a potential support [1-11].

Among tendril-bearer plants, pea plants (*Pisum sativum* L. from now on *P. sativum*) are the most studied at the genetic, morphological, physiological, and behavioral levels [2,4-20]. *P. sativum* plant development consists of different growth stages (e.g., seed germination, leaf development, flowering) aimed to increase the size and height of the plant’s body [2,4-8,11]. The pea plant’s structure is characterized by asymmetrical development of leaves along the stem. Each leaf is formed by basal, foliaceous stipules, proximal leaflets, and tendrils (Figure 1) [2,4-8,11]. All leaves (i.e., *L*_*n*_; Figure 1) show helical movements (i.e., circumnutation) along their central axis [2,11]. This movement pattern allows climbing plants to explore the environment and, once a potential support is perceived, to orient the movement of their tendrils towards it [1-20]. An unsolved issue is how the plant assigns to a specific leaf (i.e., *L*_*last*_; Figure 1) the task of veering toward and grasping a potential support. The present study aims to shed light on this matter by characterizing the behavior of the apex and the tendrils at each leaf from the seed’s germination until the support coiling using three-dimensional (3D) kinematical analysis. A control condition in which *P. sativum* plants grew in an environment lacking potential support was also considered (Figure 1). We hypothesize a kind of interactive trial-and-error process during growth. Through circumnutation at different leaves, plants may accumulate information passed from one leaf to another until the command to veer toward the support is given to the designated leaf (i.e., *L*_*last*_; Figure 1).

## Results

2

### Qualitative Results

2.1

The apex and the tendrils showed a growing pattern characterized by circumnutation. Different behavioral patterns can be observed for each leaf (Figure 1): the first two leaves (i.e., *L*_*1*_ and *L*_*2*_; Figure 1) show a pattern of growth which is primarily vertical and directed toward the light source with no development of tendrils or, at the most, just one short tendril per leaf. From the third leaf onwards (i.e., *L*_*3*_ to *L*_*last*_; Figure 1), tendrils are consistently developed, and circumnutation is more evident. Differences emerge at the level of the last leaf (i.e., *L*_*last*_; Figure 1) depending on the presence/absence of the support. For the ‘No Support’ condition, at a certain point, the plant stopped searching and collapsed (Figure 1). For the ‘Support’ condition, once the plant perceived the presence of the support, it started to bend toward it, and the tendrils assumed a choreography adequate to grasp it (Figure 1; please refer to Supplementary Materials and Supplementary Materials).

### Kinematical Results

2.2

#### Question 1 (Q1)—Is Kinematics Affected by the Presence of a Support?

2.2.1

We started our investigation by asking whether there is a specific moment at which the behavior of the plants is affected by the presence of a potential support in the environment. Plants’ growth was divided into two phases: a ‘PRE’ phase, considering the first two leaves (i.e., *L*_*1*_ and *L*_*2*_; Figure 1), and a ‘POST’ phase, considering the third leaf onwards (i.e., from *L*_*3*_ to *L*_*last*_; Figure 1). The separation between the ‘PRE’ and the ‘POST’ phase was set at 5000 min (i.e., 83.3 h–3.5 days), which equates to the temporal passage from the second to the third leaf. Mixed effects linear models with random effects (i.e., lmer) revealed a significant main effect of phase for the four kinematical indexes related to the apex, namely, the average velocity and acceleration profile of circumnutation, the distance from the origin of the plant to the center of the circumnutation and the area of circumnutation (Supplementary Materials, e.g., the velocity of circumnutation: χ^2^ (1) = 8.61, *p* = 0.003). When considering the apex, a significant interaction between the experimental condition (‘Support’ and ‘No support’) and experimental phase (‘PRE’ and ‘POST’; e.g., velocity: χ^2^ (1) = 52.05, *p* < 0.001; Supplementary Materials) was observed. Moving from the ‘PRE’ to the ‘POST’ phase, the apex exhibits a higher amplitude of average velocity (Figure 2) and acceleration (Figure 2), a longer horizontal distance from the origin of the plant to the center of the circumnutation (Figure 2), and a wider circumnutation (Figure 2). Further information is retrievable in the Supplementary Material (Supplementary Materials).

*Post hoc* analyses suggested that greater differences were evident for the ‘Support’ than for the ‘No Support’ condition (Supplementary Materials; Figure 2, e.g., velocity, Support PRE—Support POST: z = −27.02, *p* < 0.001; No Support PRE—No Support POST: z = −11.18, *p* < 0.001). Notably, differences appear to be confined to the ‘POST’ phase (e.g., velocity, Support POST—No Support POST: z = 2.93, *p* = 0.02), whereas no significant differences were observed across conditions during the ‘PRE’ phase (e.g., velocity, Support PRE—No Support POST: z = 0.88, *p* = 0.81). Importantly, higher individual variability is observed for the ‘Support’ than for the ‘No Support’ condition (Figure 2).

### Question 2 (Q2)—Is There a Benchmark in Plant Development Dynamics?

2.2.2

By inspecting the non-linear relations across dependent measures, no consistent or distinguishable trends were observed between the ‘Support’ and the ‘No Support’ conditions for the two growth phases (‘PRE’ and ‘POST’ phases; Figure 3 and Supplementary Materials). These findings suggest that pea seedlings present a similar pattern of growth, especially in the ‘PRE’ phase (i.e., *L*_*1*_ to *L*_*2*_; Figure 1), regardless of the presence of a support. The pattern of growth is characterized by a remarkable increase in terms of the average velocity and amplitude of circumnutation during the ‘POST’ phase (i.e., from *L*_*3*_ to *L*_*last*_; Figure 1), progressively showing a more complex non-linear relationship across the kinematical indices throughout the growing phase. This is evidenced by significantly higher effective degrees of freedom (i.e., edf) between indices when fitting generative additive models (i.e., gam) for each index (e.g., Figure 3, Supplementary Materials; velocity and area of circumnutation: edf = 8.19, F = 633.15, *p* < 0.001). Therefore, it may be reasonable to assume that the beginning of the third leaf (i.e., *L*_*3*_; Figure 1) may be considered a benchmark in plant development dynamics, leading to a kinematical reorganization characterized by a wider and faster search of potential support.

### Question 3 (Q3)—Is the Growth Pattern for the Apex and the Tendrils Similar across Leaf Development?

2.2.3

We explored whether the growth pattern relative to the apex was constant throughout the development of the last three leaves (e.g., *L*_*3*_, *L*_*4*_, *L*_*last*_; Figure 1) and if their growing pattern was comparable to the one exhibited by the tendrils. When inspecting the four kinematical indices, a potential coordinated growing pattern between the apex and tendrils emerges. The coordinated pattern effect appears particularly evident throughout leaves for the ‘Support’ condition (Supplementary Materials, e.g., distance from the origin of the plant to the center of the circumnutation: χ^2^ (11) = 1207.36, *p* < 0.001). The average velocity of circumnutation is described in Figure 4 for representative purposes. The description of the pattern of velocity, acceleration, and the area of circumnutation is reported within the Supplementary Materials (Supplementary Materials; Supplementary Materials). In light of this possible cross-talk among the apex and the tendrils, we reasoned that something might occur in terms of the exchange and accumulation of information that makes the plant ready to grasp when a potential support is present.

### Question 4 (Q4)—At Which Growing Stage Does the Plant Deliver the Attachment Phase?

2.2.4

We progressed by investigating variations in the number of circumnutations and switches during the development of the last three leaves (e.g., *L*_*3*_, *L*_*4*_, *L*_*last*_; Figure 1). The results showed a significant reduction in circumnutations and switches moving towards the last leaf (Supplementary Materials; circumnutations: χ^2^ (2) = 191.10, *p* < 0.001; switches: χ^2^ (2) = 31.74, *p* < 0.001). This effect was particularly evident for the ‘Support’ condition (Supplementary Materials; circumnutations: χ^2^ (2) = 11.33, *p* = 0.003, *post hoc*: Support Last Leaf—No Support Last Leaf: z = *−*4.28, *p* = 0.002; switches: χ^2^ (2) = 13.06, *p* < 0.001, *post hoc*: Support Last Leaf—No Support Last Leaf: z = *−* 7.62, *p* < 0.001; Figure 5). Overall, this pattern of results indicates that, via an iterative process, the plants reach the necessary state to perform the final launch towards the support. This is achieved by increasing the speed of circumnutation, which, in turn, corresponds to a greater area of rotation and a consistent reduction in the number of circumnutations and switch events.

When inspecting the movement time for the last three leaves developed (e.g., *L*_*3*_, *L*_*4*_, *L*_*last*_: Figure 1), the results showed that there is a gradual increase passing from the ‘Third last’ to the ‘Last’ leaf only for the ‘Support’ condition (e.g., χ^2^ (2) = 7.58, *p* = 0.022; Figure 6 and Supplementary Materials). The increase in the movement time for the last leaf may serve to implement corrective adjustments to reduce possible errors in establishing contact points for the clasping of the support [13,14].

## Discussion

3

The present study investigated how *P. sativum* plants develop ascent and attachment behavior toward a potential support. Results showed that this mechanism is associated with an iterative process related to the speed of circumnutation, which may serve as a mechanism to accumulate the necessary energy and resources for clasping a support. Note that energy here is meant to be chemical, not kinetic. The movement of the pea plant is very slow and controlled, and its mass is relatively small. Therefore, it is more likely that its movement is powered by chemical energy. As the speed of movement increases, more chemical energy is utilized. The supposed iterative process concerns both stability and plasticity. Stability is exemplified by the organizing structure of the ‘PRE’ phase (Figure 1). Here, the energy is used to increase the growth of the stem vertically towards the light source and to reach stability in the plants’ posture. Plasticity is exemplified by changes in the kinematical features (e.g., average velocity and area of circumnutation) at each leaf during the ‘POST’ phase (Figure 1). Once the plants reach stability in their posture, they invest in exploratory strategies to detect potential environmental support.

As the number of leaves developed progresses (Figure 1), the average velocity, the distance from the origin of the plant to the center of the circumnutation, the time taken to perform a circumnutation, and the area of circumnutation increases. This corresponds to a decrease in the number of circumnutations and directional switches, which reaches its peak during the last leaf (i.e., *L*_*last*_; Figure 1). This progression and changes in the features of the exploratory movements may allow plants to (i) accumulate the necessary energy and resources to implement the final launch toward the support, (ii) adjust the movement of the new leaf developed as a function of the previous and (iii) minimize errors in the vicinity of the support. In this perspective, the development of an approach-to-grasp movement in pea plants seems based on a trial-and-error process driven by an iterative algorithm (i.e., repeating a certain process several times to achieve a desired result). Exploratory movements of each leaf seem to be performed in a loop. They are constantly updated until the plant reaches the optimal solution (i.e., the final launch towards the support; convergence criterion). This is a fundamental method of problem solving characterized by repeated attempts that are performed until successful by a variety of organisms [21-23]. One avenue to be explored is Iterative Learning Control (ILC) [24], a control method for improving tracking performance in systems that repeat a given task repeatedly (i.e., trial). The basic idea behind ILC is that the information obtained from the previous trial is used to improve and correct the control input for the next trial. As the iteration continues, the system learns the task and follows the desired trajectory, minimizing possible errors [24,25]. In this way, plants might use past control information such as input signals and tracking errors to develop a successful and controlled clasping movement towards the support [26,27]. In this view, developing a functional clasping movement may entail a recurrent learning process based on a continuous cycle of information exchange between the environment and the plant.

But what kind of mechanism is responsible for ‘evaluating’ the incoming information? A likely candidate might be a cross-talk between the above- and belowground plant organs (e.g., the root system and the stem). The stem and the tendrils may acquire information through proprioception [28-31], allowing plants to perceive their position in space during movement. Feedback from this continuous proprioceptive sampling might be matched with information acquired by the root system via (i) the emission of root exudates, a cock-tail of chemical signals emitted by the roots that allow plants to explore the underground environment [32], or (ii) mechanical stimulation [11] with the roots touching the below-ground part of the support. The oscillatory movements of the roots are the result of a controlled growth in which the geotropic positive response (i.e., the directional growth of the roots with the force of gravity) is corrected by a feedback mechanism [33,34]. This system (i.e., geocontrol system; [33,34]) is regulated by: (i) the ability of the roots to perceive and process spatial information in their environment (i.e., geoperception), (ii) the conduction of this information in form of signals from the roots’ apex (i.e., the control centre) to the region of maximum elongation, and (iii) the velocity of corrective adjustment of the direction of growth. Put simply, deflections from the straight geotropic direction of growth are transmitted to the root tip, where changes in the organ’s position in relation to gravity are perceived, and from there, corrective signals are sent back to the zone of elongation by means of both chemical and electrical signals [33,34]. Importantly, it has been shown that when the root tip is removed the channel of feedback impulses to the elongation zone is interrupted [33]. In light of this, the root tips touching the belowground part of the support could generate signals that are sent to the elongation zone providing relevant information such as where the support is located. These claims are supported by a study which showed that when the support is lifted to the ground and is therefore unavailable to the root system, pea plants cannot localize it and adapt the kinematics of their approaching and grasping movement properly [16]. In light of this, things should work as follows: roots acquire information from the belowground surroundings integrated with the proprioceptive information provided by the circumnutating pattern. In the case of an unsuccessful clasping attempt, the information is sent back to the roots, which, in turn, adjusts the command for the following leaf. This iterative process continues until the information to grasp is perfected. The exchange of information from different plant sectors may occur through short- and long-distance electrical signaling processes [35-37]. Each single cell is interconnected by modular bioelectrical activities, allowing for a constant exchange of information and resources within the whole plant. The sum of bioelectrical activity (i.e., electrome) [35-37] determines a complex network signaling at the whole plant level and generates new internal schemes and changes based on environmental fluctuations, ensuring the flexible adaptation of the plant’s behavior to the surroundings [35-37]. Another possibility may rely on the propagation of chemical signals such as the growth hormones (e.g., auxin, cytokinin) through the xylem (i.e., the plant vascular tissue that conveys water and dissolved minerals from the roots to the rest of the plant) and the phloem (i.e., the plant tissue that conducts sugars from the leaves to the other parts of the plant) [38-40]. Hormone propagations are responsible for maintaining the plant’s nutritional and physical quality and for developing and growing the new organs in the plant’s above- and belowground parts of the plant [40-42]. Further, they regulate plant growth in speed and direction of movement (e.g., active bending of the organs) [31,40-42].

To conclude, the present findings strongly suggest that implementing a proper attachment plan in pea plants involves some form of adaptation driven by an iterative algorithm. Here, we set the foundation for future research, blending kinematical information with computational and physiological methods to shed more definite light on a phenomenon on the basis of a climber’s survival.

## Materials and Methods

4

### Subjects

4.1

Twenty-four snow peas (*Pisum sativum* var. saccharatum ‘Carouby de Maussane’) were chosen as the study plants (see Table 1). *P. sativum* seeds were potted and kept in the conditions outlined below.

### Experimental Conditions

4.2

*P. sativum* plants were tested in an environment in the presence (i.e., ‘Support’ condition) or in the absence of potential support (i.e., ‘No Support’ condition; Figure 7). The support was a 60 cm high wooden pole (i.e., the inground part was 7 cm long, while the aboveground part was 53 cm in height) positioned at 12 cm from the plant’s first unifoliate leaf (Figure 7).

### Germination and Growth Conditions

4.3

Cylindrical pots (diameter 20 cm; height 20 cm) were filled with silica sand (type 16SS, dimension 0.8/1.2 mm, weight 1.4). The pots were watered and fertilized using a half-strength solution culture (Murashige and Skoog Basal Salt Micronutrient Solution, SIGMA Life Science, Milan, Italy; 10×, liquid, plant cell culture tested; SIGMA Life Science, Milan, Italy) and then watered with tap water as needed three times a week. Seeds were soaked in water for 24 h and then placed in absorbent paper for 5 days to germinate. Once the seeds germinated, healthy seedlings of similar heights were chosen and potted. Each pot was then enclosed in a growth chamber (Cultibox SG combi 80 × 80 × 160 cm, Growtent, Warszawa, Poland; Figure 7) so that the seeds could germinate and grow in controlled environmental conditions. The chamber air temperature was set at 26 °C; the extractor fan was equipped with a thermo-regulator (TT125; 125 mm diameter; max 280 MC/H Vents, Kyiv, Ukraine), and there was an input-ventilation fan (Blauberg Tubo 100–102 m^3^/h, Munich, Germany). The two fan combinations allowed for a steady air flow rate into the growth chamber with a mean air residence time of 60 s. The fan was placed so air movement did not affect the plants’ movements. Plants were grown with an 11.25-h photoperiod (i.e., 5.45 a.m. to 5 p.m.) under a cool white LED lamp (V–TAC innovative LED lighting, VT–911–100 W, Des Moines, IA, USA or 100 W Samsung UFO 145lm/W—LIFUD, Suwon, Republic of Korea) that was positioned 50 cm above each seedling. Photosynthetic Photon Flux Density at 50 cm under the lamp in correspondence of the seedling was 350 umolph/(m^2^s) (quantum sensor LI–190R, Lincoln, NE, USA). Reflective Mylar^®^ (Chester, PA, USA) film of chamber walls allowed for better uniformity in light distribution. The experimental methodology was applied to the single plants grown individually in a growing chamber.

### Video Recording and Data Analysis

4.4

For each growth chamber, a pair of RGB–infrared cameras (i.e., IP 2.1 Mpx outdoor varifocal IR 1080P, Lorex, Markham, ON, Canada) were placed 110 cm above the ground, spaced at 45 cm to record stereo images of the plant. The cameras were connected via Ethernet cables to a 10-port wireless router (i.e., D–link Dsr–250n) connected via Wi-Fi to a PC, and the frame acquisition and saving process were controlled by Cam Recorder V1.0.0 software (Ab. Acus s.r.l., Milan, Italy). To maximize the contrast between the anatomical landmarks of the *P. sativum* plants (e.g., the tendrils) and the background, black felt velvet was fixed on some sectors of the walls of the boxes, and the wooden supports were darkened with charcoal. Each camera’s intrinsic, extrinsic, and lens distortion parameters were estimated using a Matlab Camera Calibrator App R2024a. Depth extraction from the single images was carried out by taking 20 pictures of a chessboard (squares with 18 mm of side, 10 columns, 7 rows) from multiple angles and distances in natural non-direct light conditions. The same chessboard used for the single camera calibration process was placed in the middle of the growth chamber for stereo calibration. The two cameras then took the photos to extract the stereo calibration parameters. In accordance with the experimental protocol, a frame was synchronously acquired every 3 min (frequency 0.0056 Hz) by the cameras. An ad hoc software (Ab. Acus s.r.l., Milan, Italy) developed by Matlab (Natick, MA, USA) was used to position the markers and track their position frame-by-frame on the images acquired by the two cameras to reconstruct the 3D trajectory of each marker [43]. All leaves developed by the plants were analyzed from the germination of the seed until the plant collapsed (i.e., ‘No Support’ condition; Figure 1) or coiled the support (i.e., ‘Support’ condition; Figure 1). For all leaves in both experimental conditions, the initial frame was defined as the frame in which the tendrils started to develop, and they were visible from the apex. The end of movement for the uncoiled leaves was defined as the frame in which the tendril(s) stopped producing their own circumnutation. For the last leaf (i.e., *L*_*last*_; Figure 1) developed by the plants, the end of movement was defined as the frame before the plant collapsed in the ‘No Support’ condition or in the ‘Support’ condition when the tendrils started to wrap around the support. The markers on the anatomical landmarks of interest of the plants, namely the apex, the junction of the tendrils, and the tendrils, were inserted *post hoc* (Figure 7). The markers were also positioned on the support (i.e., on both the lowest and the highest points of the support), the origin of the plant, and internodes as reference points (Figure 7). The tracking procedures were first performed automatically throughout the time course of the movement sequence using the Kanade–Lucas–Tomasi (KLT) algorithm on the frames acquired by each camera after distortion removal [41]. The tracking was manually verified by the experimenter, who checked the position of the frame of the marker frame by frame. The 3D trajectory [43] of each tracked marker was computed by triangulating the 2D trajectories obtained from the two cameras (Figure 7).

### Dependent Measures

4.5

The dependent variables specifically tailored to test our topic based on previous evidence [12-17,19,20,43] were: (i)Spatial trajectories: this measure allows us to describe circumnutation in both qualitative and quantitative terms.(ii)Movement time (min): The interval between the movement’s beginning and end. That is, when the plant encountered the potential support (i.e., ‘Support’ condition) or collapsed (i.e., ‘No Support’ condition).(iii)The average circumnutation velocity (mm/min).(iv)The distance from the origin of the plant to the center of circumnutation: Euclidean distance (mm) between the Circumnutation Center (i.e., the geometric center of gravity in the X–Z plane computed as the mean of each coordinate for all the points constituting the circumnutation) and the plant origin in the X–Z plane.(v)Area of the circumnutation (mm^2^) as the sum of pixels with a value equal to 1, obtained from the binarization of the circumnutation trajectory.(vi)Average circumnutation acceleration (mm/min^2^).(vii)Number of circumnutations performed by the plant during the entire movement time.(viii)Number of switch directions during each circumnutation (clockwise, counterclockwise, and none). For each circumnutation, the sum of all the angles between the movement vector at time t and the movement vector at time t + 1 is calculated. The direction, then, is determined according to the following logic: if the resulting sum is equal to 2π *±* 1.2, then the direction is counterclockwise, or else if the resulting sum is equal to −2π *±* 1.2, then the direction is clockwise. For all other cases, no direction is assigned.

The kinematical indices were scaled for standardization purposes.

### Statistical Analysis

4.6

Data analyses were computed in the R environment [44]. Categorical variables, such as experimental phase (‘PRE’, ‘POST’) or leaf (‘Third last’, ‘Second last’, ‘Last’), were created for analysis. For the investigation of Q1 (see Section 2 section), four mixed-effects linear models (i.e., lmer; ‘lme4’) [45] were fitted for each kinematical variable, setting the interaction between experimental condition (‘Support’, ‘No Support’) and experimental phase, as well as random intercept (plants, 24 specimens) and random slope (experimental condition). The apex of each plant retrieved data, and the total number of observations considered for each model was equal to 3117. Dependent variables were scaled for standardization purposes. For answering Q2, four Generalized Additive Models (i.e., gam) from the ‘mgcv’ R package [46] were fitted (with the REML method) for the investigation of non-linear relationships between each of the four kinematical variables and the other indices, controlling for experimental condition and experimental phase. Predictors were smoothed to study the complexity of the non-linear relationship with the dependent variables, which were scaled as well as the predictors. The smoothed random intercept of each plant was considered (n = 24). Data were retrieved by the apex of each plant, and the total number of observations considered for each model was equal to 3115. The Supplementary Materials represent relationships among kinematical indices by fitting B-spline curves with three degrees of freedom. For the investigation of Q3, four mixed-effects linear models (‘lme4’) [45] were fitted for each kinematical variable, testing the interaction between experimental condition, leaf, and the anatomical landmark of the plant (‘Apex’, ‘Tendrils’), also setting random intercept (plants, 24 specimens) and random slope (experimental condition). Data were retrieved by the apex and tendrils (as a single object) of each plant, and the total number of observations considered for each model was equal to 5895. Again, dependent variables were scaled. To investigate Q4, we fitted a mixed-effects linear model to test the interaction between experimental condition and leaf on variations in the total number of circumnutations, while also setting plants as random intercepts. Data were retrieved by the apex and tendrils (as a single object) of each plant, and the total number of observations considered for each model was equal to 478. Contextually, a generalized mixed-effect linear model (i.e., glmer) was fitted to the data to investigate the interaction between experimental condition, leaf, and switch direction (i.e., ‘Clockwise’, ‘Counterclockwise’) on variations in the total number of switches, setting a Poisson family distribution and plants as random intercepts. The present analysis did not consider switches with undetermined directions. Finally, movement time was investigated by fitting a mixed-effect linear model, setting the ‘Movement time’ of the leaves dependent on the interaction between the leaf and experimental condition. The apex of each plant retrieved data, and the total number of observations considered for each model was equal to 3117. Considering the main interest in the interaction effects, Type 3 Sum of Squares was considered for deriving statistical results from lmer and glmer models. (R package ‘car’) [47]. *Post hoc* analyses were computed through the pairwise contrast test of the ‘emmeans’ R package [48] when needed. For descriptive purposes, relationships among each kinematical index and experimental time are represented by fitting B-spline curves with three degrees of freedom. Descriptive graphics and model plots were developed via ‘ggplot2’ R package [49] and are retrievable in the Supplementary Materials.

## Supplementary Material

Supplementary material

Video S1

Video S2

## Figures and Tables

**Figure 1 F1:**
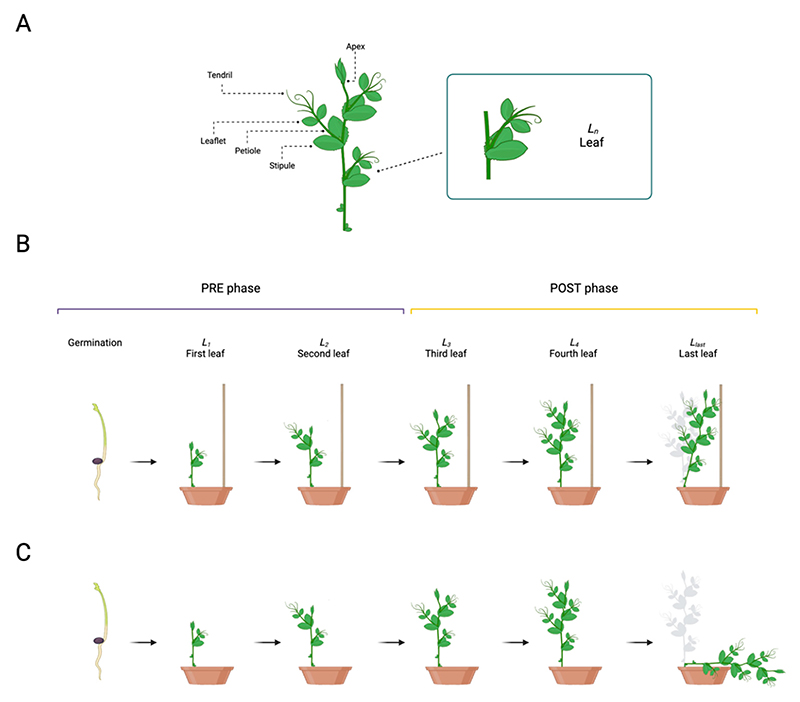
(**A**) Graphical representation of the morphological structure of the leaf consisting of the stipule, petiole, leaflet, and tendril(s). (**B,C**) Graphical representations of the development of different pea plant leaves from the seed germination until the clasping of the support (i.e., ‘Support’ condition) or the falling of the plant (i.e., ‘No Support’ condition). Plant growth was divided into two phases: the ‘PRE’ (i.e., from the germination to the second leaf; *L*_*2*_) and the ‘POST’ phases (i.e., from the third leaf to the last leaf developed; *L*_*3*_ to *L*_*last*_).

**Figure 2 F2:**
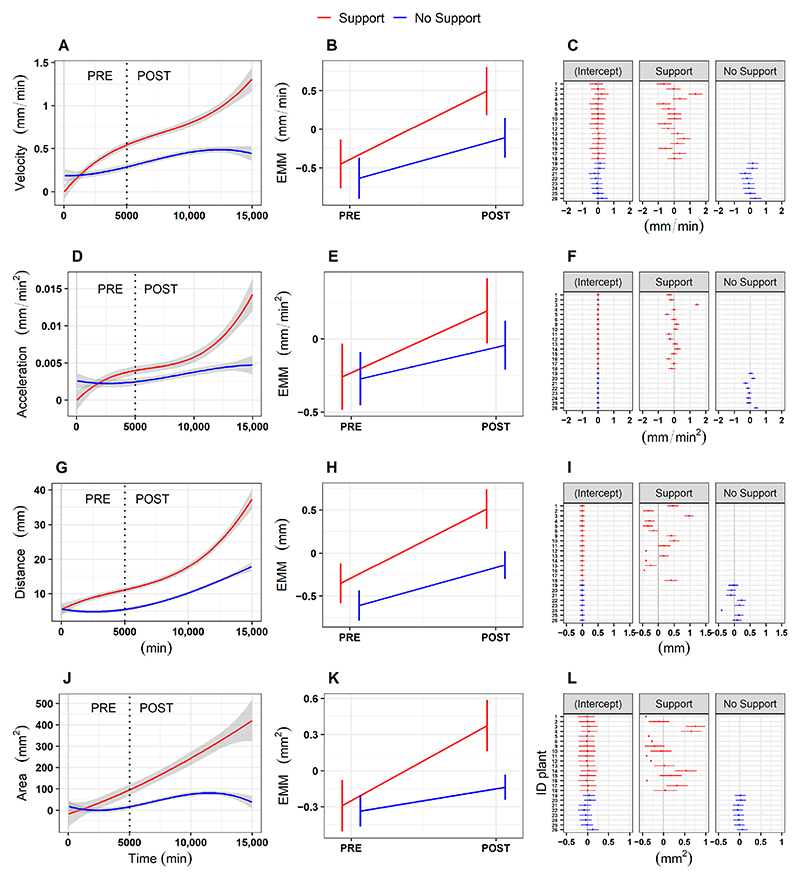
(**A,D,G,J**) Graphical representation of average velocity and acceleration profiles of the circumnutation, the distance from the origin of the plant to the center of the circumnutation, and the area of the circumnutation for the apex in the ‘Support’ and the ‘No support’ conditions. (**B,E,H,K**) Graphical representation of *post hoc* analysis for the interaction between experimental condition (‘Support’, ‘No Support’) and experimental phase (‘PRE’, ‘POST’) for the four kinematical variables. (**C,F,I,L**) Graphical representation of estimated values for each plant as a random intercept and random slope of the fitted models, controlling for the experimental condition (‘Support’, ‘No Support’) for the four kinematical variables. The ‘Support’ condition is represented with a red line, and the ‘No Support’ condition with a blue line. EMM = Estimated marginal means.

**Figure 3 F3:**
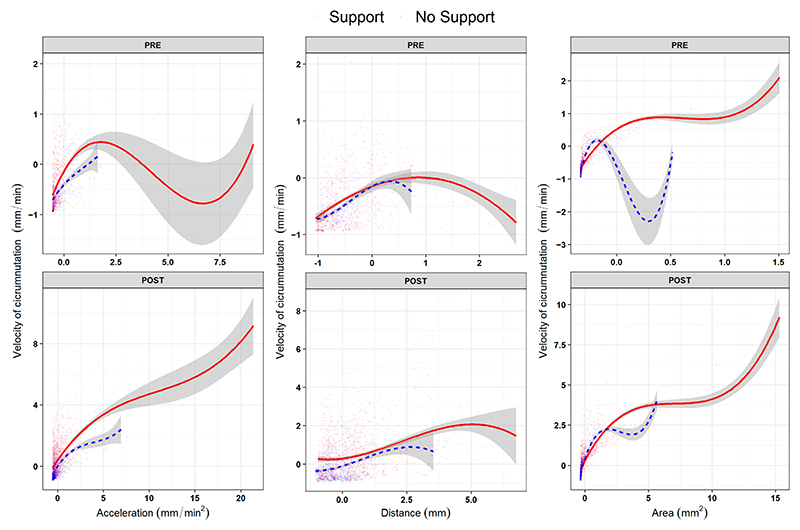
B-spline curves (degrees of freedom = 3) for the non-linear relationship across experimental phases (row facets ‘PRE’, ‘POST’) between the average velocity of circumnutation (as the scaled dependent variable, y axes) and the other three kinematical variables (column facets): acceleration of circumnutation, distance from the origin of the plant to the center of circumnutation and area of circumnutation. Data represent the sole activity of the apex. The ‘Support’ condition is represented by the red solid line, and the ‘No Support’ condition is represented by the blue dashed line.

**Figure 4 F4:**
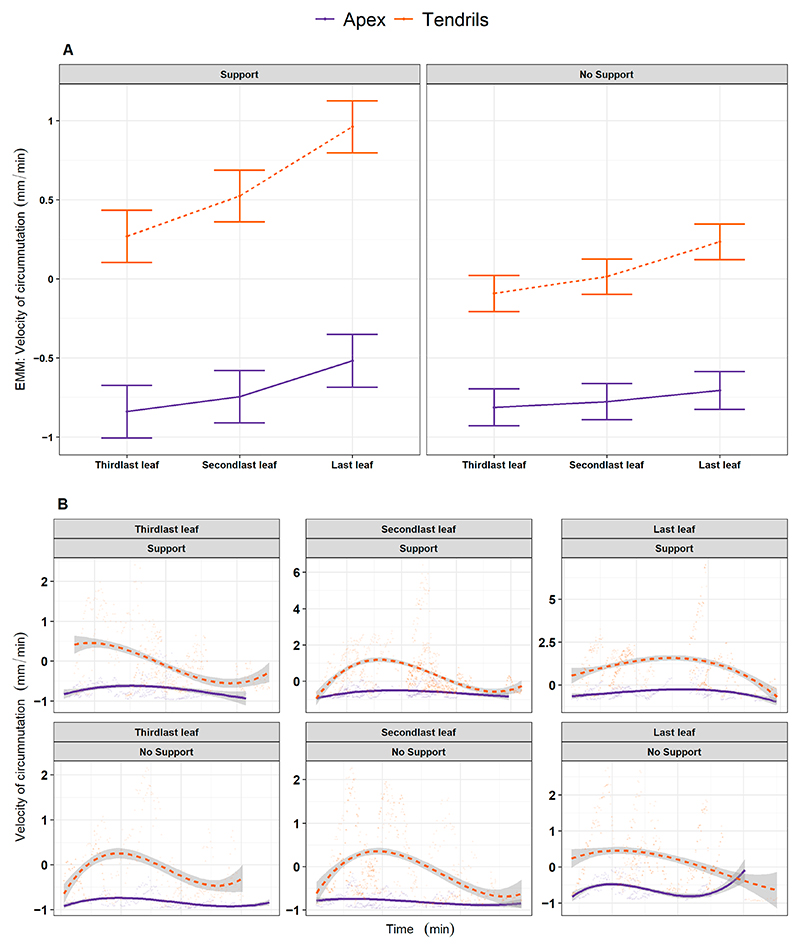
(**A**) Graphical representation of *post hoc* analysis for the interaction between experimental condition (‘Stimulus’, ‘No Stimulus’), anatomical landmark of the plant (‘Apex’, ‘Tendrils’) and leaf (‘Third last’, ‘Second last’, ‘Last’) for the estimation of the average velocity of circumnutation (scaled). (**B**) For descriptive purposes, the distribution of the same kinematical variable is represented as smoothed across the three last leaves of interest, controlling for the same experimental factors. Tendrils are represented by the orange dash line. The apex is represented by the violet solid line.

**Figure 5 F5:**
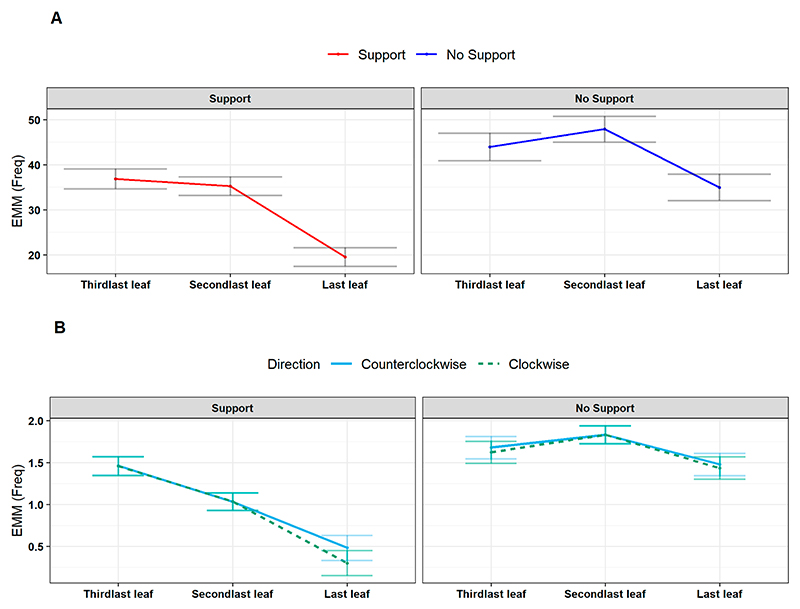
(**A**) Estimated marginal means (EMM) of the number of total circumnutations controlling for experimental condition (in facets, ‘Support’, ‘No Support’) and leaf (‘Third last’, ‘Second last’, ‘Last’). The ‘Support’ condition is represented by the red line, and the blue line represents the ‘No Support’ condition. (**B**) Estimated marginal means of total switches controlling for the experimental condition (in facets, ‘Support’ and ‘No Support’) and leaf (‘Third last’, ‘Second last’, ‘Last’) and for the switch direction (‘Clockwise’, ‘Counterclockwise’). The dark green represents the clockwise switch direction dashed line and the counterclockwise with the light blue solid line.

**Figure 6 F6:**
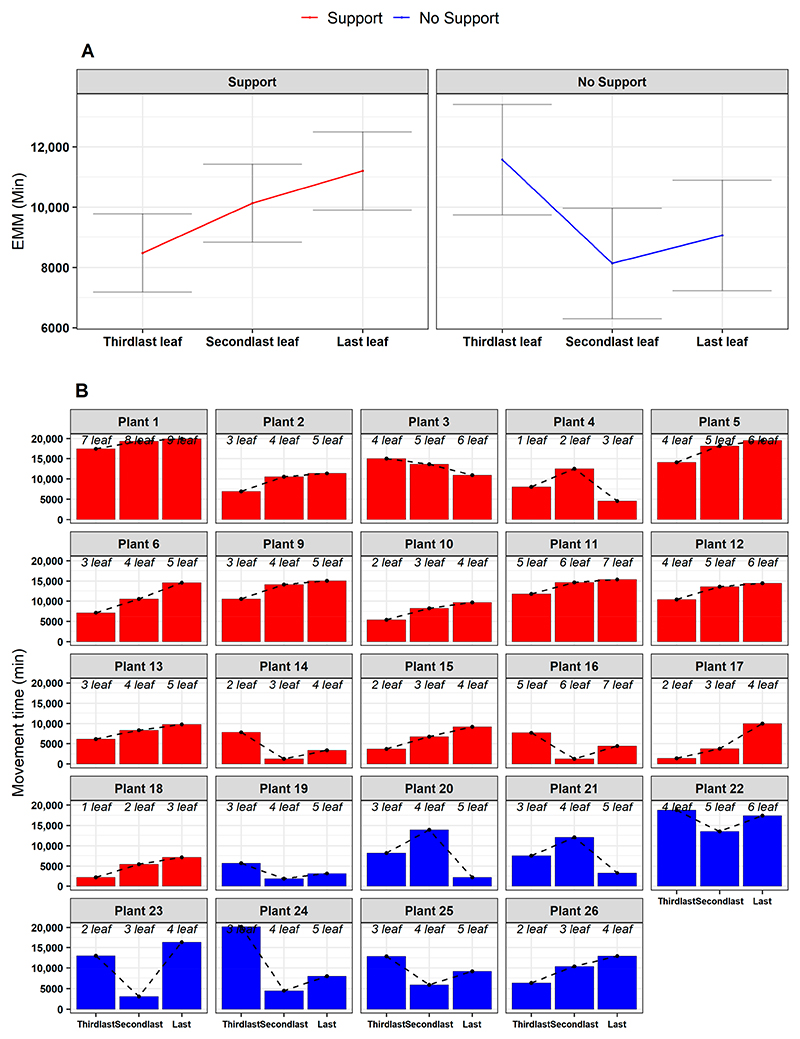
(**A**) Estimated marginal means of the movement time for leaf, controlling for the experimental condition (‘Support’, ‘No Support’). The ‘Support’ condition is represented in red, and the ‘No Support’ condition in blue. EMM = Estimated marginal means. (**B**) Frequency bar plots representing the total movement time of the last three leaves developed (in columns) per each plant (in facets), controlling for experimental condition (‘Support’, ‘No Support’). The dotted lines further describe the variation in the movement time of leaves. Inside each plot, italic text indicates the correspondence between leaf (‘Third last’, ‘Second last’, ‘Last’) and absolute movement time for each plant.

**Figure 7 F7:**
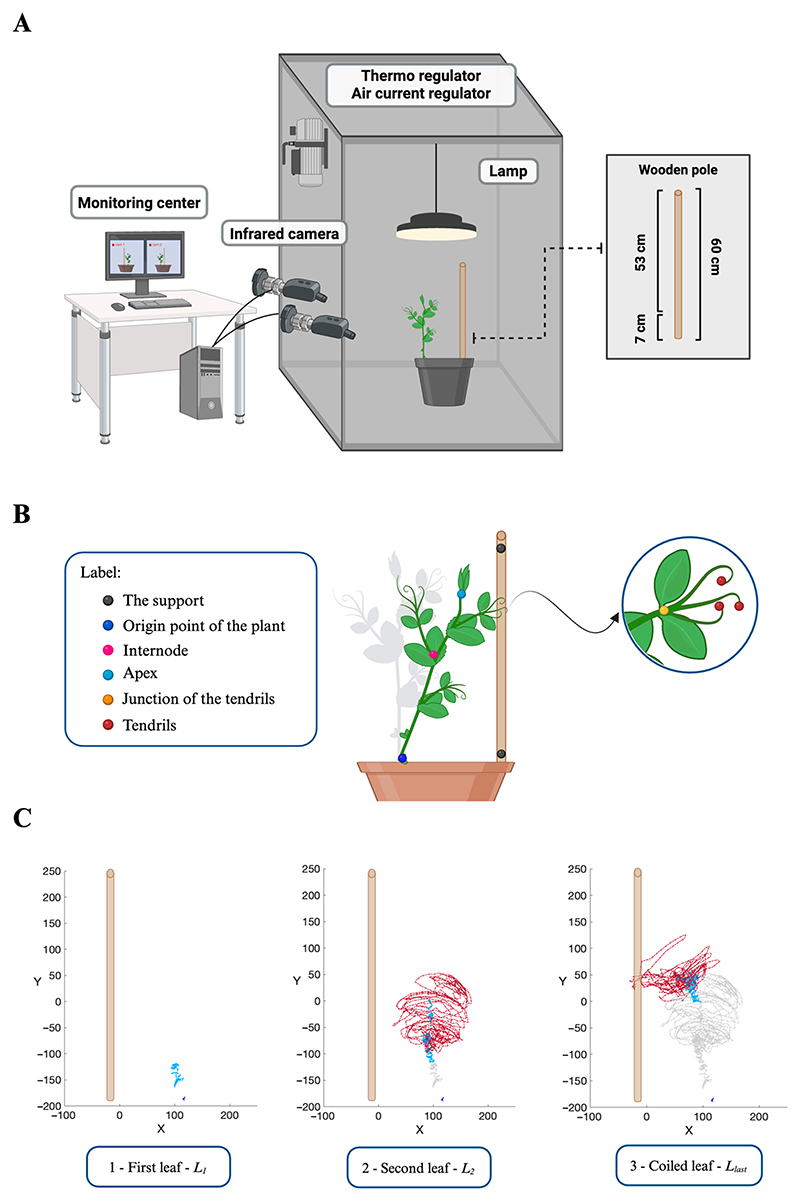
(**A**) Graphical illustration of the experimental setup and how *P. sativum* plants were potted together with potential support (i.e., ‘Support’ condition). Each chamber was equipped with two infrared cameras on one side, a thermoregulator for temperature control, two fans for input and output ventilation, and a lamp positioned upon the pot at a distance of 50 cm. The potential support was a wooden pole with a height of 60 cm and a diameter of 1.2 cm, which was positioned at a distance of 12 cm in front of the first leaf of the plant. (**B**) Graphical illustration of the anatomical landmarks of the plants and the reference points considered in the kinematical analysis. (**C**) Representative trajectories for each leaf developed in the ‘Support’ condition. For the ‘Support’ conditions, the last leaf veers toward the support, and the tendrils grasp the support. x = x-axis; y = y-axis; red dashed line = tendril; light blue dashed line = apex; blue dashed line = origin point of the plant.

**Table 1 T1:** Sample description.

	No Support condition	
N˚		8
Distance		–
Age		21.12d (±1.5)
N˚ of leaves		5 (±0.93)
	Support condition	
N˚		16
Distance		12 cm
Age		23d (±1)
N˚ of leaves		5 (±1.67)

Note. The age of the plant was expressed in days, and the N^*°*^ of leaves refers to the mean, while the standard deviation is noted in parentheses.

## Data Availability

The original data presented in the study are openly available in Zenodo at: https://doi.org/10.5281/zenodo.11091797.

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
