# Peer review of "Ascent and Attachment in Pea Plants: A Matter of Iteration"

_plants, 2024, doi:10.3390/plants13101389_

Round 1

Reviewer 1 Report

Comments and Suggestions for Authors

Title: ascent and attachment in pea plant a matter of iteration

Authors : Guerra et al.,

General comment The authors explored a fundamental question in hoe plants with tendrils find their support, through a number of cleverly designed fundamental research questions and a very simple experimental design with a complex data analysis. The paper is a delight to read and the authors do a remarkable job to present it. There are no main issues at the conceptual, data collection and analysis levels, as I see it.

Minor points.

Here and there the authors refer to plants in a matter that alludes to being sentient. A careful review of the manuscript should be taken to remove that assumption.

For instance line 39 “how the plant decide to assign”…here decide assumes a level of decision and consciousness which is indeed not the case. Remove decide leaving “how the plant assigns”…

Line 32, few extra …. To be removed

Post hoc; ie are Latin-derived words therefore should be Italicised. See lines, 36, 95, 102 etc.

Author Response

Minor points.

1. Here and there the authors refer to plants in a matter that alludes to being sentient. A careful review of the manuscript should be taken to remove that assumption. For instance line 39 “how the plant decide to assign”…here decide assumes a level of decision and consciousness which is indeed not the case. Remove decide leaving “how the plant assigns”…

R1. We thank the Reviewer for bringing this issue to our attention. As recommended, we have now removed the term ‘decide’ in line 39, p. 1 and carefully reviewed the main text.

2. Line 32, few extra …. To be removed

R2. The extra dots have been removed (line 32 p. 1).

3. Post hoc; ie are Latin-derived words therefore should be Italicised. See lines, 36, 95, 102 etc.

R3. We have italicized the Latin words throughout the text in the new version of the manuscript.

Reviewer 2 Report

Comments and Suggestions for Authors

This is very carefully planned and performed study reporting important new results. I have only one remark on the potential role of root apices in control of shoot/tendril movements. In the reference 16, authors reported that pea plants grasped the support only if roots were intacts as well as only if the support was inserted into the soil up to level of root apices. This is important issues and should be discussed too. In this respect, it would be relevant to mention pioneering studies of Milos Spurny (Spurny 1968, 1973).

Spurny M. (1968) Effect of root tip amputation on spiral oscillations of the growing hypocotyl with radicle of the pea (Pisum sativum L.). Biologia Plantarum 10: 98-111

Spurny M. (1973) Parameters of spiral oscillations as indicating the efficiency of control system of growing roots. Biologia Plantarum 15: 358-360

Author Response

1. This is very carefully planned and performed study reporting important new results. I have only one remark on the potential role of root apices in control of shoot/tendril movements. In the reference 16, authors reported that pea plants grasped the support only if roots were intacts as well as only if the support was inserted into the soil up to level of root apices. This is important issues and should be discussed too. In this respect, it would be relevant to mention pioneering studies of Milos Spurny (Spurny 1968, 1973).

Spurny M. (1968) Effect of root tip amputation on spiral oscillations of the growing hypocotyl with radicle of the pea (Pisum sativum L.). Biologia Plantarum 10: 98-111

Spurny M. (1973) Parameters of spiral oscillations as indicating the efficiency of control system of growing roots. Biologia Plantarum 15: 358-360

R1. We thank the Reviewer for raising this important issue and pointing towards this literature. As suggested, we have now integrated the manuscript by reporting the studies of Spurny (1968, 1973). Please refer to lines 248-265 in the new version of the manuscript.